# *PIK3R1*^W624R^ Is an Actionable Mutation in High Grade Serous Ovarian Carcinoma

**DOI:** 10.3390/cells9020442

**Published:** 2020-02-14

**Authors:** Concetta D’Ambrosio, Jessica Erriquez, Maddalena Arigoni, Sonia Capellero, Gloria Mittica, Eleonora Ghisoni, Fulvio Borella, Dionyssios Katsaros, Silvana Privitera, Marisa Ribotta, Elena Maldi, Giovanna Di Nardo, Enrico Berrino, Tiziana Venesio, Riccardo Ponzone, Marco Vaira, Douglas Hall, Mercedes Jimenez-Linan, Anna L. Paterson, Raffaele A. Calogero, James D. Brenton, Giorgio Valabrega, Maria Flavia Di Renzo, Martina Olivero

**Affiliations:** 1Candiolo Cancer Institute, FPO-IRCCS, Candiolo, 10060 Torino, Italy; concetta.dambrosio@unito.it (C.D.); jessica.erriquez@ircc.it (J.E.); sonia.capellero@ircc.it (S.C.); mittica@aslvco.it (G.M.); eleonora.ghisoni@ircc.it (E.G.); elena.maldi@ircc.it (E.M.); enrico.berrino@ircc.it (E.B.); tiziana.venesio@ircc.it (T.V.); riccardo.ponzone@ircc.it (R.P.); marco.vaira@ircc.it (M.V.); giorgio.valabrega@ircc.it (G.V.); martina.olivero@unito.it (M.O.); 2Department of Oncology, University of Torino, Candiolo, 10060 Torino, Italy; 3Department of Molecular Biotechnology and Health Sciences, University of Torino, 10126 Torino, Italy; maddalena.arigoni@unito.it (M.A.); raffaele.calogero@unito.it (R.A.C.); 4Città della Salute e della Scienza, 10126 Torino, Italy; fulvio.borella87@gmail.com (F.B.); d.katsaros@libero.it (D.K.); sprivitera@cittadellasalute.to.it (S.P.); mribotta@cittadellasalute.to.it (M.R.); 5Department of Life Sciences and Systems Biology, University of Torino, 10125 Torino, Italy; giovanna.dinardo@unito.it; 6Department of Medical Sciences, University of Torino, 10126 Torino, Italy; 7University of Cambridge, Cambridge CB2 0XZ, UK; Douglas.Hall@cruk.cam.ac.uk (D.H.); mercedes.jimenez-linan@addenbrookes.nhs.uk (M.J.-L.); alp37@cam.ac.uk (A.L.P.); James.Brenton@cruk.cam.ac.uk (J.D.B.); 8Cancer Research UK Cambridge Institute, Cambridge CB2 0RE, UK

**Keywords:** ovarian cancer, PI3K, PIK3R1, Patient-Derived xenografts, PDX derived tumour cells

## Abstract

Identifying cancer drivers and actionable mutations is critical for precision oncology. In epithelial ovarian cancer (EOC) the majority of mutations lack biological or clinical validation. We fully characterized 43 lines of Patient-Derived Xenografts (PDXs) and performed copy number analysis and whole exome sequencing of 12 lines derived from naïve, high grade EOCs. Pyrosequencing allowed quantifying mutations in the source tumours. Drug response was assayed on PDX Derived Tumour Cells (PDTCs) and in vivo on PDXs. We identified a *PIK3R1*^W624R^ variant in PDXs from a high grade serous EOC. Allele frequencies of *PIK3R1*^W624R^ in all the passaged PDXs and in samples of the source tumour suggested that it was truncal and thus possibly a driver mutation. After inconclusive results in silico analyses, PDTCs and PDXs allowed the showing actionability of *PIK3R1*^W624R^ and addiction of *PIK3R1*^W624R^ carrying cells to inhibitors of the PI3K/AKT/mTOR pathway. It is noteworthy that *PIK3R1* encodes the p85α regulatory subunit of PI3K, that is very rarely mutated in EOC. The *PIK3R1*^W624R^ mutation is located in the cSH2 domain of the p85α that has never been involved in oncogenesis. These data show that patient-derived models are irreplaceable in their role of unveiling unpredicted driver and actionable variants in advanced ovarian cancer.

## 1. Introduction

Epithelial Ovarian Cancer (EOC) is a heterogeneous disease with five major histologic types [1]. The most frequent is high grade serous EOC (HGS-EOC), characterized by disruption of TP53 [2]. The majority of patients with HGS-EOC present with advanced stage disease and there has been little improvement in overall survival with standard treatment, which has not changed over the past 20 years and relies on cytoreductive surgery and platinum-based combination chemotherapy [3]. Maintenance therapy with inhibitors of the Poly (ADP-ribose) polymerase (PARP) now offers significant survival advantages for patients with BRCA1 and BRCA2 mutations [4,5], but targeted therapy for other HGS-EOC patients is much less developed compared to other solid tumours. In melanoma and non-small cell lung carcinomas, specific genetic alterations have been identified as key oncogenic drivers, i.e., as mutations able to confer a selective advantage to a cell, through either increasing its survival or proliferation and, thus, able to cause clonal expansion of the carrier cell. These findings have led to the development of therapies targeting these mutations, which has provided demonstrable clinical benefit [6]. Thus, a number of these mutations have also been defined as “actionable”, not only because their functional outcome makes carrier cells responsive to a targeted therapy, but also thanks to the availability of a specific targeted drug.

In ovarian cancer, a handful of aberrations in cancer genes have been found at high frequency, but many more cancer-related genes are found mutated at a very low frequency (≈1%) [7,8]. This so-called “long-tail” of candidate driver mutations [9] is still incompletely characterized not only in ovarian cancer but also across other human cancers [10]. Many of these mutations are not even well characterized oncogenic drivers [7,8] and evidence of the potential clinical impact is lacking.

Thus, the goal of this study is to identify driver and therapeutically actionable molecular alterations in ovarian cancer. Primarily, with this purpose in mind, we [11] and other groups [12,13,14,15,16,17,18] have developed platforms of Patient-Derived Xenografts (PDXs) of EOC that can recapitulate the tumour biology and clinical responses observed in donor patients. PDXs provide the most conclusive tool to validate low frequency mutations as biomarkers for targeted therapy, as evidenced by the use of colorectal cancer PDX models to predict successful clinical trials [19].

Using PDX and PDX Derived Tumour Cells (PDTC), we provide here the identification and the detailed functional and therapeutic validation of a *PIK3R1*^W624R^ mutation found in HGS-EOC. In HGS-EOC point, mutations of genes encoding proteins of the PI3K-AKT-mTOR signalling pathway are very rare, whereas this pathway is very frequently affected by large genetic aberrations. The importance of this pathway in this and other cancer histotypes has prompted the clinical development of >40 different inhibitors [20]. However, we show here that although rare, the *PIK3R1*^W624R^ is likely important, because it is identified as a truncal mutation in both the PDX line and source tumour. Thus, the PDX model has been instrumental in propelling the analysis, precisely because the genomic evolution of the source tumour grown in the immunodeficient animal model [21] did not counter-select this mutation. Moreover, we show here that the PDX model has been invaluable for functional validation, as it allowed overcoming questionable assays in test tubes or in unrelated cell types.

## 2. Materials and Methods

### 2.1. Cell Lines

The OVCAR-8 (NCI-DTP Cat# OVCAR-8) were obtained from the NCI-DCTD repository. The A2780 (NCI-DTP Cat# A2780) ovarian carcinoma cells lines and the LNCaP prostate carcinoma cell line (CLS Cat# 300265/p761_LNCaP) were purchased from American Type Culture Collection (ATCC,). Cell cultures were maintained following the protocols suggested by the providers.

### 2.2. Patient-Derived Xenograft (PDX) Platform

Between 2012 and 2016, a collection of 167 separate clinical samples of epithelial ovarian cancers was undertaken, with the understanding and written consent of each subject before they participated in the study, and according to the PROFILING Protocol, and it also conformed to the standards set by the Declaration of Helsinki, and was approved by the Regione Piemonte Ethical Committee (approval number 5141 on 9/3/2011), and then by the IRCCS Ethical Committee (approval number 192/2016 on 19/7/2016).

Samples have been implanted intra-peritoneum and subcutaneously in the right flank of severely immunocompromised NOD/Shi-scid/IL-2R γnull mice and subsequently propagated subcutaneously, based on our and other authors’ data showing that the continued development of the PDX model subcutis was suitable [11,12]. Sample quality was assessed by the pathologists. A total of 65 PDX lines successfully grew (thus with an approximate take of 40%) and were propagated for at least three passages. Appendix A shows the full characterization of tumours and a quality assurance of 43/65 PDX lines, carried out in accordance with Meehan et al. [22] within the first three passages of PDX lines, which resulted in loss of human stroma. For treatment, animals were randomized using the Laboratory Assistant Suite [23]. Animal treatment was initiated when mean xenograft volume was approximately 100 mm^3^. Buparlisib diluted in N-Methyl-2-pyrrolidone and PEG 10/90 *v*/*v* was administered by oral gavage every day 5 days/week for 3 weeks. Buparlisib was administered at the dose of 20 mg/kg, that was reported to be effective in diverse PDX models [24]. Treatment with buparlisib at doses >30 mg/kg resulted in mouse toxicity. Control animals were treated with vehicle alone. All animal procedures were approved by the local Ethical Commission and by the Italian Ministry of Health in accordance with EU Directive 2010/63/EU for animal experiments; a first authorization was obtained on 12/7/2012 and, following subsequent regulations, approved again on 14/01/2016 (no. 16/2016-PR) and extended for two additional years on 17/9/2018. All animal procedures comply with the “3R” principles. Additional methodological details of animal experimentation are reported in Appendix A. The completed ARRIVE guideline checklist for reporting experiments using live animals is attached as Appendix A.

### 2.3. TMAs and IHC

Tissue Microarray (TMA) preparations and staining were carried out as described by Sapino et al. [25] TMA slides for the characterization of the PDX lines were stained with the Ventana automated immunostainer (BenchMark AutoStainer, Ventana Medical Systems, Tucson, AZ, USA) using the following antibodies by Ventana Medical Systems: anti-WT1 Cat# 760-4397, anti-p53 Cat# 790-2912, anti-EPCAM Cat# 760-4383, anti-Cytokeratin 7 Cat# 790-4462 and anti-CD20 Cat# 760-2531, the latter to rule out the growth of lymphoma, which occurred in 10–20% of cases, as also reported by others [26]. Immunohistochemical detection of S6 and P-S6 in TMA slides of PDXs was carried out using rabbit monoclonal antibodies (Cell Signalling Technology; Denvers, MA, USA) Cat# 2217 and Cat# 4858, respectively, and anti-rabbit Ig (K4003) and EnVision system purchased from Dako (Agilent, Santa Clara, CA, USA). The immunohistochemical detection of Ki67 and P-S6 in sections of PDXs was carried out using mouse monoclonal (Dako clone MIB-1) and rabbit monoclonal antibodies (Cell Signalling Technology Cat# 4858) respectively, anti-mouse or anti-rabbit Ig (K4001 and K4003, respectively) and EnVision system purchased from Dako (Agilent, Santa Clara, CA, USA). Quantification of P-S6 and Ki67 positive cells was carried out using the Colour Deconvolution plug-in in ImageJ in 20–30 and 10–15 separated fields, respectively, of each PDX.

### 2.4. PDX Derived Tumour Cell (PDTC) Preparation

PDX samples have been chopped and digested with a Human Tumour Dissociation Kit (Miltenyi Biotec, Bergisch Gladbach, Germany) according to the manufacturer protocol. Human cells were isolated using a Mouse Cell Depletion Kit (Miltenyi Biotec). Cells were plated, and after 24–48 h treated with a drug in 96-multiwell plates.

### 2.5. Viability and Cytotoxicity Assays

A CellTiter-Glo^®^ assay was used to evaluate the viability of PDTCs and cell lines after 72 h treatment with drugs, according to the manufacturer’s protocol (Promega, Madison, WI, USA). Drugs were purchased from Selleck Chemicals (Houston, TX, USA). GR values have been calculated for each concentration as reported in Hafner et al. [27] and plotted using a GraphPad Prism version 7.02 (San Diego, CA, USA).

A c ytotoxicity assay was carried out as follows. Seventy-two hours after treatment, cells in 96-well plates were fixed with 2% paraformaldehyde in PBS for 40 min, washed twice in PBS and stained with 10% crystal violet in 20% methanol for 40 min. Plates were washed extensively and lysed in 10% acetic acid. The absorbance was measured at 595 nm using a microplate reader (BioTek Synergy HTX, Winooski, VT, USA).

### 2.6. Western Blot Analysis

PDTCs were treated for 24 h with the indicated drug or the vehicle 24 h after plating. From PDTCs and PDX samples, proteins were extracted in ice cold elution buffer (TrisHCl pH 7.4, containing EDTA, 1% Triton X-100, 10% glycerol and protease and phosphatase inhibitors). Proteins of snap frozen PDXs have been extracted as above after GentleMacs (Miltenyi Biotec) digestion. Western blot (WB) analysis was carried using the following antibodies: rabbit polyclonal anti-AKT (Cell Signalling Technology Cat# 9272), rabbit monoclonal Phospho(Ser473)-AKT (Cell Signaling Technology Cat# 4060) and polyclonal goat anti-vinculin (N-19) (Santa Cruz Biotechnology Cat# sc-7649). Labelled secondary antibodies have been revealed with ECL (Thermo Fisher Scientific, Waltham, MA, USA) using the ChemiDoc Touch Imaging System (BioRad, Hercules, CA, USA).

### 2.7. Crystal Structure Analysis

The position of W624 residue in the structure of human p85α encoded by *PIK3R1* was predicted through sequence alignments and structure superimposition. Sequence alignments and domain assignment were performed using PSI-Blast (NCBI BLAST), whereas structure superimposition was performed using UCSF Chimera.

### 2.8. WES and CNA Analysis

Exome library preparation was performed using the Nextera Rapid Capture Enrichment kit from Illumina. Genomic DNAs were quantified using the Qubit system (Invitrogen, Carlsbad, CA, USA) and 50 ng were used as input material for library preparation. Pools of 12-plex libraries (500 ng each library) were hybridized with capture probes (coding exomes oligos) twice: a first hybridization step for 2 h and a second for 18 h. After elution and clean up, enriched DNA libraries were amplified with 10-cycle PCR. The products were purified, loaded on a bioanalyser using DNA 1000 chip for quality control and quantified with Qubit. Samples were sequenced using the NextSeq 500 platform (Illumina, San Diego, CA, USA) as paired 150 bp reads, using the NextSeq 500/550 High Output Kit v2, loading 1.4 pM DNA and obtaining as cluster density an average of 200 K/mm^2^ clusters. Microarray HumanCytoSNP for CNA analysis was performed by Genomix4life S.R.L. (Baronissi, Salerno, Italy), using 200 ng of each DNA hybridized for 18 h at 48 °C on HumanCytoSNP-12 v2.1 BeadChip, according to the manufacturer’s instructions, and analysed with an Illumina iSCAN. WES and SNP data analyses were performed using the pipeline shown in Appendix A, using each patient’s germline DNA as reference. The data generated or analysed during this study are included in this article, or if absent, are available from the corresponding author upon reasonable request.

### 2.9. Pyrosequencing Analysis

The presence and allele frequency of *TP53* and *PIK3R1* mutations in source tumours of any PDX line were assessed using pyrosequencing (QIAGEN, Hilden, Germany). Briefly, sample DNAs were amplified, and the PCR products were subjected to the ‘sequence by synthesis’ pyrosequencing method following the manufacturer’s instructions, using the primers and conditions reported in Appendix A. Results were analysed with the PSQ24 software for allelic quantification.

## 3. Results

### 3.1. Mutational Burden of PDX Lines from HGS-EOCs

Forty-three PDX lines derived from epithelial ovarian carcinomas were fully characterized according to the minimal information standard for reporting PDX data [22], including clinical and pathological attributes of the patient’s tumour, the processes of implantation and passaging in the host mouse strain, and quality assurance by means of genotypic and phenotypic characterization. The information is reported in Appendix A. Briefly, we compared the histology of each PDX to that of the source tumour and classified histotype using immunohistochemistry (IHC) with CK7, EPCAM, WT1 and p53 antibodies, which confirmed the diagnosis of high grade serous histology in 25/43 lines. Figure 1 shows the histology and IHC staining of TMAs of representative PDX lines and the corresponding source tumours. Targeted NGS of *TP53* showed that all the PDXs from HGS-EOC harboured *TP53* mutations (Appendix A). *TP53* aberrations were protein truncating or missense mutations previously shown to be pathogenic according to the IARC *TP53* Database (http://p53.iarc.fr/). The detailed list of aberrations is reported in Appendix A. Samples with missense mutations in *TP53* had evidence of nuclear p53 stabilization as indicated by high protein expression in tumours and xenografts (Appendix A). The PDX line with null splice site mutation did not express p53 protein. The targeted NGS of BRCA1/2 showed mutations of these genes classified as likely pathogenic according to the BRCA Mutation Database (http://arup.utah.edu/database/BRCA) in 19% of the above PDX lines, i.e., close to the expected frequency. The likely impactful aberrations and the variants of unknown significance detected in the 43 PDX lines are listed in Appendix A.

We then further studied 12 PDX lines propagated from treatment naïve HGS-EOCs, listed in Appendix A, that also reports clinical information.

In the 12 PDX lines, using Whole Exome Sequencing (WES), we detected a total of 2743 single nucleotide variants (SNVs) in 2314 genes. The majority (79%) of mutated genes in our analysis have been reported as mutated in the TCGA analysis of 523 HGS-EOC [8]. Most (438/487) of the mutated genes not listed in the TCGA analyses of HGS-EOC have been found mutated in other cancer histotypes, according to the Pan-Cancer Atlas.

*TP53* was the only gene mutated in all 12 PDX lines, and the only gene found mutated across more than one PDX line (Appendix A), with an allele frequency of approximately = 1, in line with occurrence of LOH. As expected, in the studied 12 PDX lines, WES detected the same *TP53* mutations identified with targeted NGS.

### 3.2. Somatic PIK3R1^W624R^ Mutation in HGS-EOC

To identify possible driver and actionable genetic aberrations in the 12 above-mentioned PDX lines, we considered SNVs and copy number alterations (CNAs) affecting cancer-related genes reported in COSMIC (CGCv84) [28].

CNAs detected in cancer genes corresponded to those reported in TCGA for HGS-EOC (Appendix A). Among genes reported in TCGA as amplified in more than 10% of cases [29], in our small series (Appendix A) we found the expected frequency of increased copy number of *CCNE1* and *PIK3CA*. LOH of the *TP53* gene in each of the PDX lines was confirmed (compare the data of Appendix A).

The list of cancer genes harbouring SNVs in the 12 PDX lines is shown in Figure 2 and Appendix A.

It came out that, in 3/12 PDX lines, WES did not detect mutations in known cancer genes other than *TP53*. A number of SNVs in cancer genes were identified as Single Nucleotide Polymorphisms (SNPs). All the variants are listed in Appendix A. Only some of the somatic mutations in the remaining 9 PDX lines were predicted to be damaging and/or deleterious based on the bioinformatics tools used for analysis (Figure 2, SIFT, PROVEAN and FATHMM [30]). Possible actionability (Figure 2) was evaluated using the Drug Gene interaction database (DGidb) [31].

In the PDX line #475, we found a point mutation of the *PIK3R1* gene that results in the W624R residue substitution in the encoded protein. This was detected with an allele frequency approximately = 1.0, which suggested that this was a loss of function mutation followed by LOH, in accordance with *PIK3R1* classification as a tumour suppressor gene. As explained above, based on the bioinformatic tools (see Figure 2), this mutation was predicted to be deleterious and damaging and possibly actionable. *PIK3R1* encodes the p85α regulatory subunit of the p110α catalytic subunit of the Phosphatidyl-Inositol-3 Kinase (PI3K). Some of the already known mutations of the p85α subunit affect the intracellular signalling of the PI3K pathway, leading to PI3K activation, via distinct molecular mechanisms [32]. Activation of the PI3K pathway in this PDX line #475 was shown by the immunohistochemical detection of the phosphorylation of the S6 protein, that is a proxy of AKT activation (Appendix A). In other PDX lines, without mutations of genes involved in the PI3K/AKT/mTOR pathway, the P-S6 protein is not comparably increased (Appendix A).

*PIK3R1*^W624R^ was detected in all the parallel and sequential passages of the PDX line #475 and in each passage with an allele frequency approximately = 1.0. In these passages also, the same *TP53* mutation was found with an allele frequency approximately = 1.0, in line with *TP53* mutation being a truncal mutation, as previously shown (see e.g., ref. [33]). Thus, data suggested that the *PIK3R1*^W624R^ might also be a truncal mutation in PDXs. Most other mutations were also consistently detected in all passages of this PDX line, whereas only a few were limited to a subset of passages. The list of public, shared and private mutations found in passages is shown in Figure 3A. The consistency of public SNVs in parallel and sequential passages demonstrated the genetic stability upon passages of the PDXs carrying the *PIK3R1*^W624R^ mutation.

The *PIK3R1*^W624R^ PDX line was propagated from a biopsy sample taken at laparoscopy from a patient with a stage IIIc HGS-EOC. This PDX line responded as well as the patient to platinum-based chemotherapy. Indeed, the patient received neo-adjuvant platinum-based chemotherapy followed by surgery. Regrettably, the patient relapsed after six months and received a second platinum-based line that came out to be limitedly effective. At post-chemotherapy surgery, another sample was propagated as the PDX line, which no longer responded to carboplatin. In this paired PDX line the *PIK3R1*^W624R^ mutation was also found with an AF approximately = 1. More importantly, we found the same *PIK3R1*^W624R^ mutation in specimens of the source tumour. Pyrosequencing of two distinct FFPE samples of the source tumour showed that in each FFPE sample the percentage of the *PIK3R1*^W624R^ sequence was the same as that of the *TP53* allele found mutated in the PDX line (panels B–E of Figure 3). The percentage of the mutated *TP53* allele was considered a reliable proxy of the percentage of tumour cells in the source tumour specimens.

### 3.3. Ex Vivo and In Vivo Assays of PIK3R1^W624R^ Actionability

As mentioned above, to estimate the therapeutic implications of any given mutation, we first used the Drug Gene interaction database (DGidb) [31] that suggested the actionability of *PIK3R1*^W624R^. Another resource, the DEPO database (Database of Evidence for Precision Oncology, http://depo-dinglab.ddns.net/), that focuses on specific mutations for therapeutic projections, was not useful because it does not report hotspot regions of the *PIK3R1* gene. However, we focused further attention on this mutation because of the potential actionability of a member of the PI3K-AKT-mTOR pathway.

Hypothesis on the functional role of any given mutation can also be formulated based on crystal structures of the molecules and complex of interest. Thus, we investigated the possible role of the W624R mutation in the interaction between the p85 isoforms and p110 isoforms of PI3K (Figure 4).

Regrettably, only the crystal structure of human p110α in complex with nSH2 and iSH2 domains of p85α is available (PDB ID 4L1B, Figure 4B). Hence, we explored the possible role of the amino-acid residue homologue to the W624 in mouse p85 being available in the crystal structure of mouse p110β isoform in complex with iSH2 and cSH2 domains of mouse p85β isoform (PDB ID 2Y3A, Figure 4C). Structure superimposition showed that mouse p85β shares common folds with the human p85α counterpart. Sequence alignments demonstrated that human p85α and mouse p85β display 73% homology and 59% identity, whereas human p110α and mouse p110β sequences have 57% homology and 40% identity. According to sequence alignment (Figure 4D), the W624 in the human p85α corresponds to W616 in the mouse cSH2 domain of p85β. These data analyses showed that the W616 in the cSH2 domain of mouse p85β is not involved in the interaction between murine p85β and p110β. Thus, the structure-based prediction was inconclusive.

Thus, we carried out the functional assays (Figure 5) of the *PIK3R1*^W624R^ in cells of the *PIK3R1*^W624R^ carrying PDX line #475 ex vivo, by challenging the susceptibility to inhibitors of the PI3K/AKT/mTOR pathway that could be activated by suppression of the p85α regulatory subunit and fruitfully targeted for therapy.

To test cell susceptibility to drugs ex vivo, we propagated the PDX Derived Tumour Cells (PDTCs) depleted of mouse cells as short-term cultures. Viability assays were carried out by exposing PDTCs to inhibitors for 72 h. The susceptibility of *PIK3R1*^W624R^ PDTCs was compared to that of the control cell lines (Figure 5) and to that of the PDTCs derived from the #2085 PDX line (Appendix A). The latter PDTCs were propagated from a bona fide HGS-EOC, carrying mutations of the *TP53* and *BRCA2* genes (Appendix A), but not of genes involved in the PI3K/AKT/mTOR pathway. The following cell lines were assayed as controls, selected based on mutation spectrum (Cancer Cell Line Encyclopedia, https://portals.broadinstitute.org/ccle) and susceptibility or resistance to any given drug: A2780 ovarian carcinoma cells, carrying *PIK3CA* activating mutation and *PTEN* loss that make them highly sensitive to the pan-class I PI3K inhibitor buparlisib (BKM120), to the p110α specific inhibitor alpelisib (BYL-719) and to the dual PI3K/mTOR inhibitor dactolisib (BEZ235); the OVCAR-8 ovarian carcinoma cells with wild-type *PIK3CA* and *TP53* gene mutation, but known to be susceptible to the dual PI3K/mTOR inhibitor dactolisib (BEZ235), as reported by the Genomic of Drug Sensitivity in Cancer Project (https://www.cancerrxgene.org/), and the *PTEN*-mutated LNCaP prostate carcinoma cells that are known to be exquisitely susceptible to the p110β specific inhibitor GSK2636771. To take into account cell division rates across the PDTCs and cell lines, growth rate inhibition metrics were used, i.e., GR-calculator [27].

The *PIK3R1*^W624R^ #475 PDTCs but not the non-mutated #2085 PDTCs, were sensitive to the pan-class I PI3K inhibitor buparlisib (BKM120) as shown in Figure 5 and Appendix A, respectively. Figure 5 also shows the susceptibility of the *PIK3R1*^W624R^ #475 PDTCs to the p110α specific inhibitor alpelisib (BYL-719), compared to control cell lines. Similarly, these #475 PDTCs showed susceptibility to the dual PI3K and mTOR inhibitors dactolisib (BEZ235). Conversely, the p110β selective inhibitor GSK2636771 did not affect the *PIK3R1*^W624R^ PDTCs. The susceptibility of *PIK3R1*^W624R^ PDTCs to the four inhibitors was confirmed with a cytotoxicity assay (Appendix A). In line with the results of viability assays, buparlisib, dactolisib and alpelisib, but not GSK2636771, affected AKT phosphorylation (Figure 5), that is a proxy of PI3K activation status.

Moreover, in vivo buparlisib delayed the growth of the *PIK3R1*^W624R^ PDXs (Figure 6A). As the total decrease of tumour volume could not allow proper estimation of possibly cytostatic agents, we demonstrated also the strong reduction of Ki67 positive, i.e., proliferating, cells in treated PDXs (Figure 7A,C). AKT phosphorylation was reduced in xenografts in line with growth delay by buparlisib treatment (Figure 6B). Accordingly, phosphorylation of the S6 protein, that is a proxy of the PI3K/AKT activation, was reduced (Figure 7B,D).

## 4. Discussion

We demonstrate here that a rare mutation in the *PIK3R1* tumour suppressor gene [29] is actionable in HGS-EOC using validated patient-derived models of the disease. *PIK3R1* is very frequently mutated in other cancer histotypes, with the notable exception of HGS-EOC [10], in particular in uterine carcinomas and carcinosarcomas, glioblastoma, breast and colorectal cancer. Most of the mutations reported so far are located in the iSH2 domain of the *PIK3R1* encoded p85α, which is known to bind and regulate the p110α subunit of PI3K [32]. The *PIK3R1*^W624R^ mutation described here is located in the cSH2 domain of the p85α. The same W624R amino acid change has been reported previously, but not characterized, in one colorectal cancer [34], while a W624C substitution has been reported in one non-small cell lung carcinoma (NSCLC) [35], and a nucleotide change in the same codon, leading to a non-sense mutation being found in one stomach cancer sample [36] and one endometrial carcinoma [37]. Other rare mutations in the cSH2 of the *PIK3R1* gene are reported in COSMIC (https://cancer.sanger.ac.uk/cosmic) and TumorPortal (http://www.tumorportal.org). The function of a number of these mutations has been studied in test tubes (see e.g., ref. [38]) or in functional assays in cell line models with totally different genetic backgrounds (see e.g., refs. [39,40]). As a result, the potential oncogenicity of only two mutations in the p85α cSH2 domain has been proposed based on the promotion of survival and growth of Ba/F3, a murine interleukin-3 dependent pro-B cell line. On the other hand, in the cSH2 domain of the p85α, several germline mutations are located, which in heterozygosity cause SHORT syndrome [41]. This syndrome is characterised by developmental defects and is historically defined by its acronym: Short stature (S), Hyperextensibility of joints and/or inguinal Hernia (H), Ocular depression (O), Rieger abnormality (R) and Teething delay (T). Molecular studies have shown that *PIK3R1* mutations associated with SHORT syndrome result in defects, and not in the activation of PI3K signalling [41,42].

Surprisingly, another mutation in the cSH2 domain (the R649W substitution) has been reported in both endometrial carcinoma and as a hotspot in four families affected by SHORT syndrome [42]. Therefore, although prediction algorithms allowed us to classify the W624R amino acid change in *PIK3R1* as possibly deleterious and damaging the protein product, it was difficult to predict its actual impact on ovarian cancer. As structure-based approaches are more specific than sequence-based approaches at predicting driver mutations [40], we endeavoured to model the association in p85α between the W624R mutation and the p110α subunit, but available crystal structures did not help in defining the impact of the *PIK3R1*^W624R^ on human PI3K function. It has also been postulated that the role of the cSH2 domain of the human p85α in oncogenesis relies on its interaction with and inhibition of the p110β isoform of the PI3K [43]. We disproved this hypothesis with functional assays ex vivo that showed directly the susceptibility in vitro of *PIK3R1*^W624R^ PTDCs to the p110α specific inhibitor alpelisib but not to the p110β specific inhibitor. Susceptibility to pathway inhibition was confirmed by treating PDTCs with a dual PI3K/mTOR inhibitor and with the pan class I PI3K inhibitor buparlisib, which also inhibited the growth of PDXs in vivo. These data confirmed the requirement of proper functional assays for evaluating the driving oncogenic effect of any given mutation.

Other observations suggest that the impact of any mutation as well as its actionability could be better studied in patient-derived models and in the actual tissue affected. Nowadays, the traditional system of anatomic cancer classification has been overcome by a classification system based on molecular alterations shared by tumours across diverse tissue types. This concept has led to the development of so-called basket or umbrella trials. However, exceptions that challenge this concept have also become apparent from such notable examples as the unpredictable clinical responses to a potent BRAF inhibitor across diverse malignancies, all expressing the same *BRAF* mutation [44].

Altogether, the data show that, to assess the function of mutant alleles, assays in patient-derived models of the relevant cancer, such as PDXs and short-term PDTCs, are extremely important. Long-term cultures of patients’ samples [45] are useful for drug screening. Short-term monolayer cultures [46] and organoids [47] derived from patients’ biopsies have been shown as a valuable model for rapid drug testing and thus for co-clinical trials, but their application may be limited by the modest take and throughput, respectively. Thus, all the patient-derived experimental models should be considered complementary and not alternative, as every model system is imperfect and suitable in its own way.

The validation of *PIK3R1*^W624R^ as a biomarker of response to PI3K/AKT/mTOR pathway inhibitors exceeds the aim of this work. The PI3K-AKT-mTOR signalling pathway is one of the most frequently dysregulated ones in human cancers. More than 40 inhibitors of this pathway have reached various stages of clinical development, but only a few have been approved by FDA and EMA [48]. In clinical studies, inhibitors of the pathway have shown limited efficacy and/or manageability [20]. P110 isoform-specific inhibitors appear more promising in trials [49,50,51]. In ovarian cancer, gene abnormalities other than homologous recombination defects are even more difficult to pair with an approved or investigational drug [52].

## 5. Conclusions

In the ovarian cancer field, suitable preclinical models such as PDXs are still invaluable for population-based studies, as they might better mimic the inter-tumour heterogeneity that is seen in patients and might be predictive of the clinical efficacy of targeted drugs. This model allowed us to identify a rare mutation in the *PIK3R1* gene in a domain that was not expected to be involved in oncogenesis, and showing that this mutation makes ovarian cancer cells responsive to inhibitors of the PI3K/AKT/mTOR pathway.

## Figures and Tables

**Figure 1 cells-09-00442-f001:**
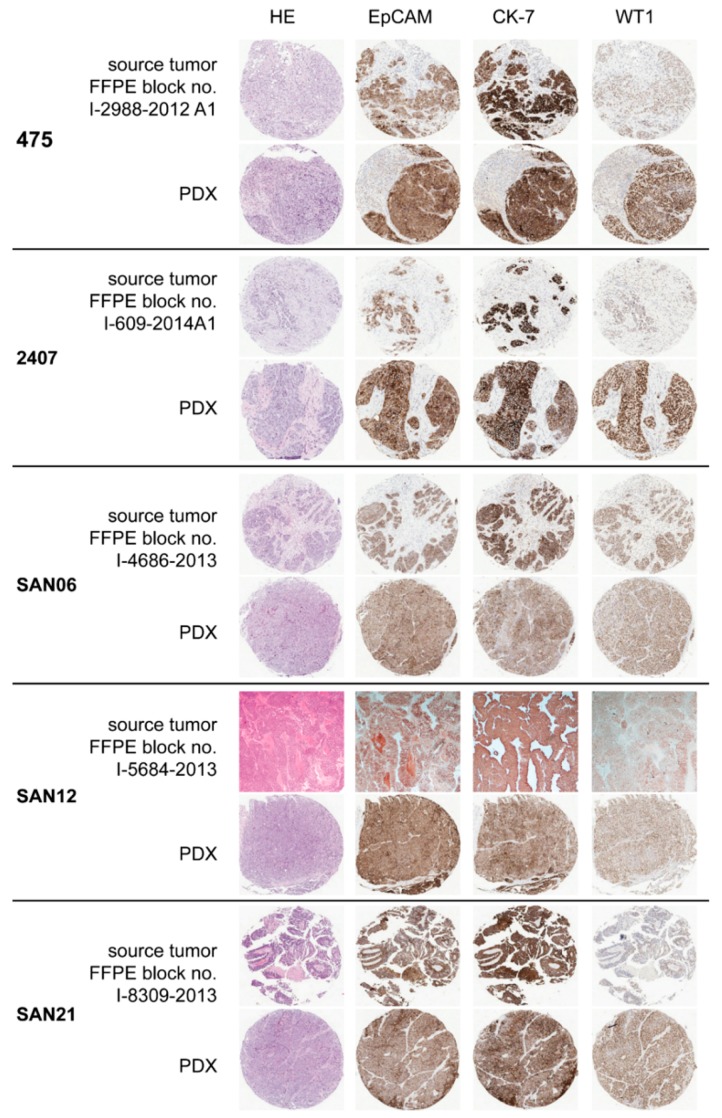
Histological characterization of PDX lines. Representative images of TMAs of PDX lines compared to sections of the corresponding source tumours. Numbers on the left are those of PDX lines as catalogued by the PROFILING approved protocol, while the numbers of the used FFPE block of samples of source tumours are shown on the top left of each panel. The complete characterization of these and the other thirty-eight PDX lines is reported in Appendix A.

**Figure 2 cells-09-00442-f002:**
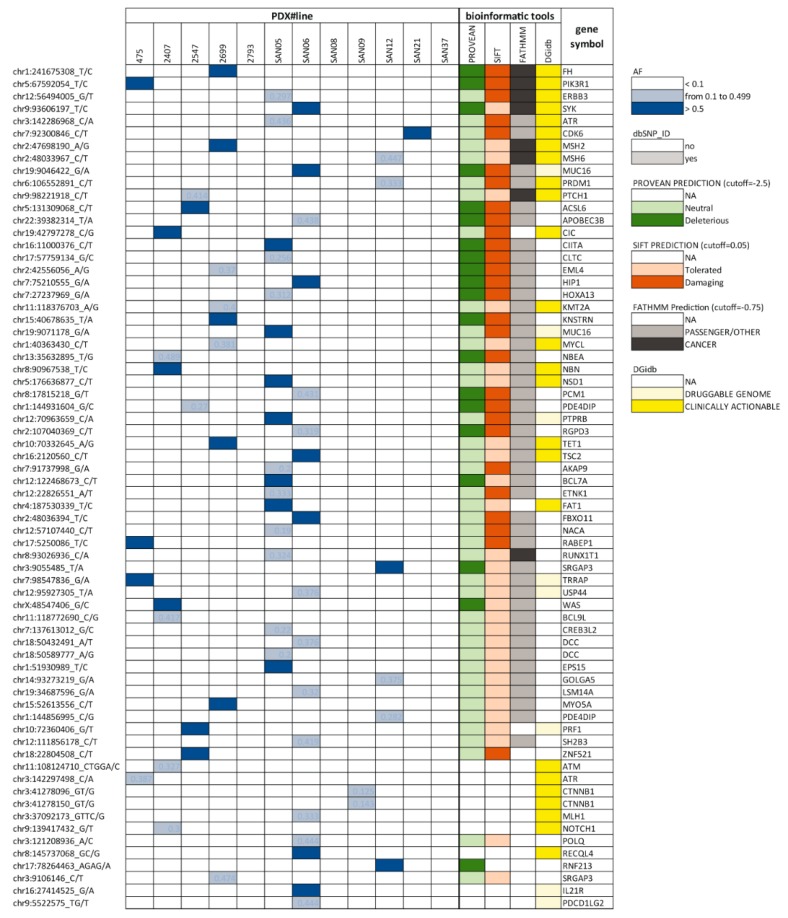
Single Nucleotide Variants (SNVs) in cancer genes found in PDX lines derived from naïve HGS-EOC. Variants with an allele frequency (AF) ≥ 0.1 are listed. Only SNVs not classified as SNPs based on the SNP database are shown in this Figure 2. All the variants, including those classified as SNPs, are reported in the related Appendix A. Legend to boxes is shown on the right.

**Figure 3 cells-09-00442-f003:**
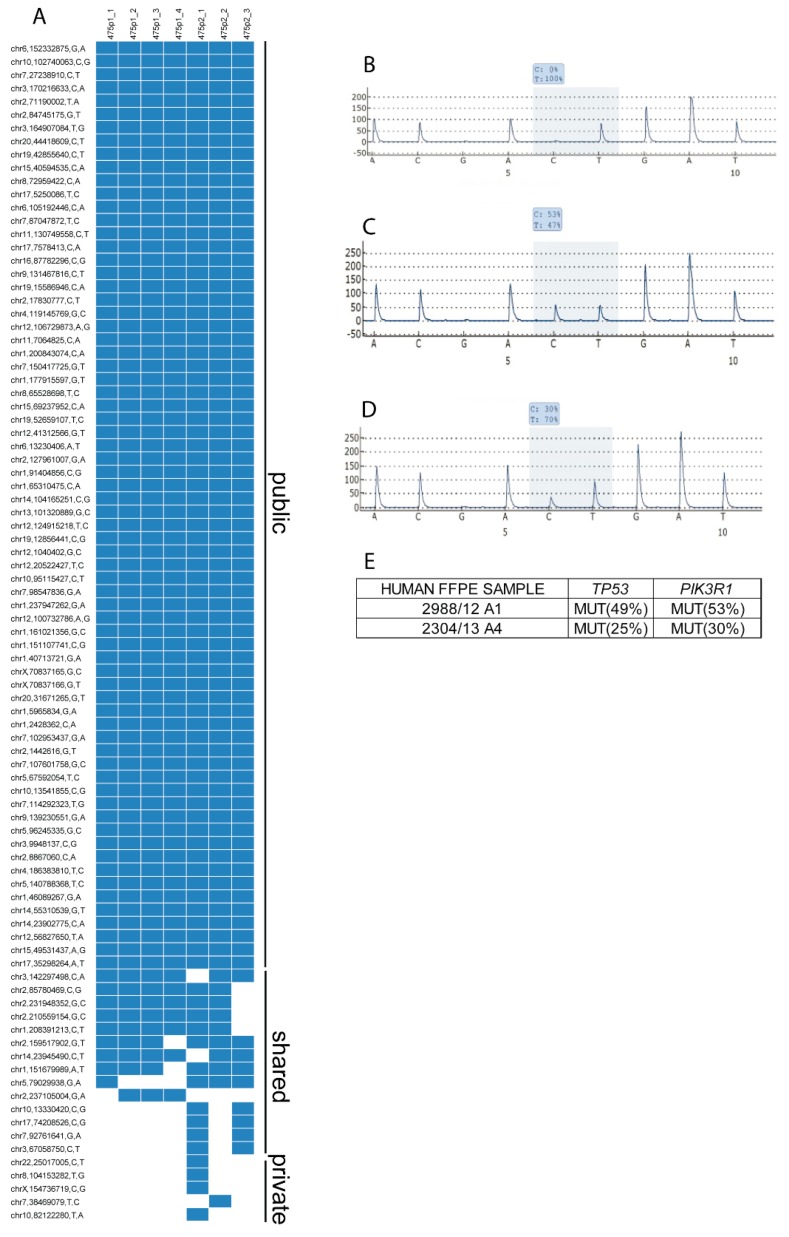
Identification of the *PIK3R1*^W624R^ mutation as truncal mutation in parallel and serial passages of the PDX line #475 and in the corresponding source tumour. (**A**) The W624R mutation in *PIK3R1* is one of several public mutations found in seven parallel and serial passages of the PDX line#475. (**B**–**E**) Pyrosequencing analysis confirmed the presence of the *TP53* and the *PIK3R1* mutations found in the #475 PDX line in two FFPE samples from distinct blocks of the source tumour. The *TP53* and the *PIK3R1* mutated sequences showed the same allele frequency (AF) in each sample. The AF of the PDX line-specific *TP53* mutation was considered as a proxy of the percentage of tumour cells in the human tumour samples. (**B**) Sequence of *PIK3R1* in Control Reference Genome; (**C**) percentage of *PIK3R1*^W624R^ in FFPE sample A1 from the paraffin block 2998 of the source tumour; (**D**) percentage of *PIK3R1*^W624R^ in FFPE sample A4 from the paraffin block 2304 of the source tumour; (**E**) percentage of mutated sequences of *TP53* and *PIK3R1* in the two above FFPE samples.

**Figure 4 cells-09-00442-f004:**
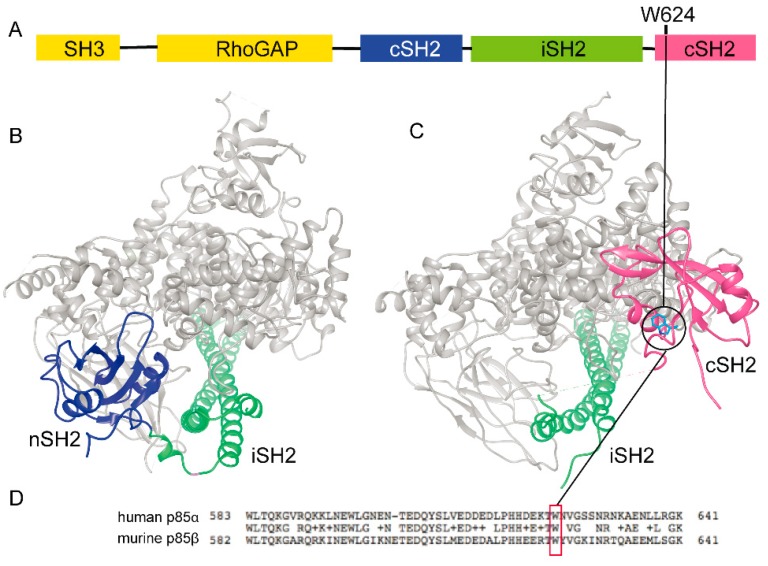
Crystal structures showing interaction of the p110 isoforms with the p85 isoforms in human and mouse PI3K. (**A**) Domain organization of p85α. (**B**) Available crystal structure (PDB ID 4L1B) of human p110α isoform with catalytic activity (grey) complexed with nSH2 (blue) and iSH2 (green) domains of human p85α; (**C**) available crystal structure (PDB ID 2Y3A) of mouse p110β in complex with iSH2 (green) and cSH2 (pink) domains of mouse p85β. (**D**) Alignment of the cSH2 domains of the human p85α and mouse p85β; homology is shown in the middle: W624 of the human p85α protein is conserved and corresponds to the W616 of the mouse p85β protein (red box in D).

**Figure 5 cells-09-00442-f005:**
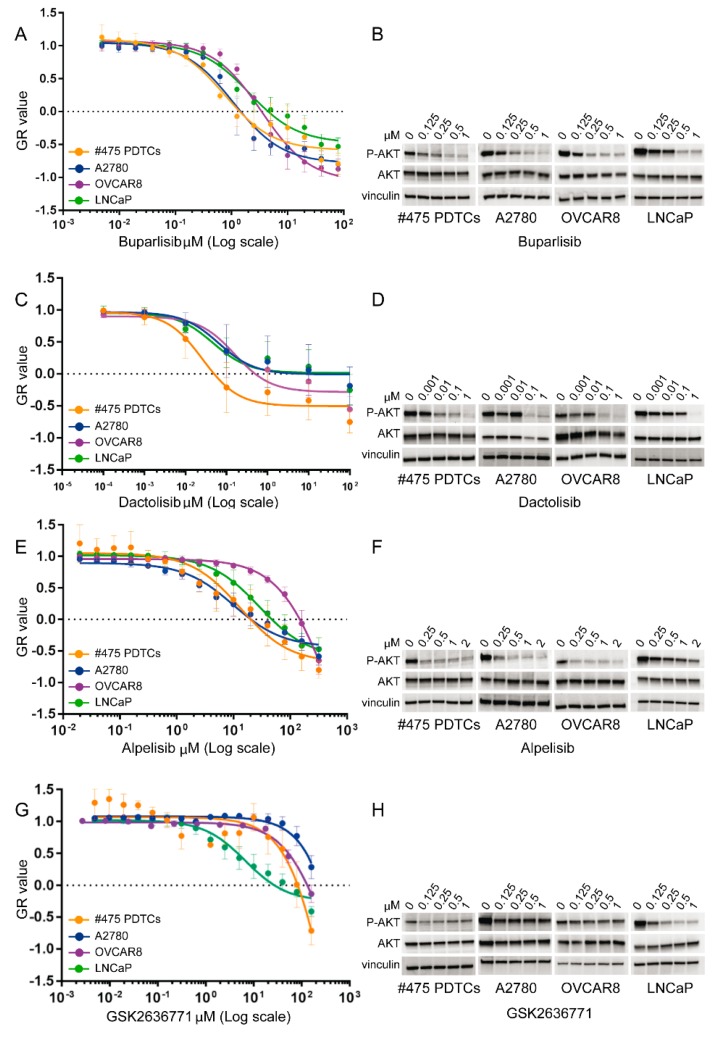
Response of *PIK3R1*^W624R^ carrying PDTCs to inhibitors of the PI3K/AKT/mTOR pathway. In each experiment, control cell lines were assayed, too. (**A**,**C**,**E**,**G**): Dose-response curves in CellTiterGlo 72 h viability assays. Normalized growth rate (GR value) inhibition metrics of three replicate experiments is shown to take into account cell division rates. The sign of GR values relates directly to response phenotype: positive for partial growth inhibition, zero for complete cytostatic effect and negative for cytotoxicity. The x axis shows drug concentration on a log_10_ (Log) scale. (**B**,**D**,**F**,**H**): Western blot analysis of the phosphorylation of the AKT signal transducer in response to drugs, as a proxy of PI3K activation status. (**A**) The *PIK3R1*^W624R^ PDTCs showed susceptibility to the pan-class I PI3K inhibitor buparlisib (BKM120), comparable to that of the highly responsive A2780 cells; (**C**) the *PIK3R1*^W624R^ PDTCs showed susceptibility to the dual PI3K/mTOR inhibitor dactolisib (BEZ235), comparable to that of the most sensitive (OVCAR-8) of the above cell lines; (**E**) the *PIK3R1*^W624R^ PDTCs were also highly susceptible to the p110α specific PI3K inhibitor alpelisib (BYL719) as well as the A2780 cells; and (**G**) resistant to the p110β specific inhibitor GSK2636771 to which the LNCaP cells are exquisitely susceptible.

**Figure 6 cells-09-00442-f006:**
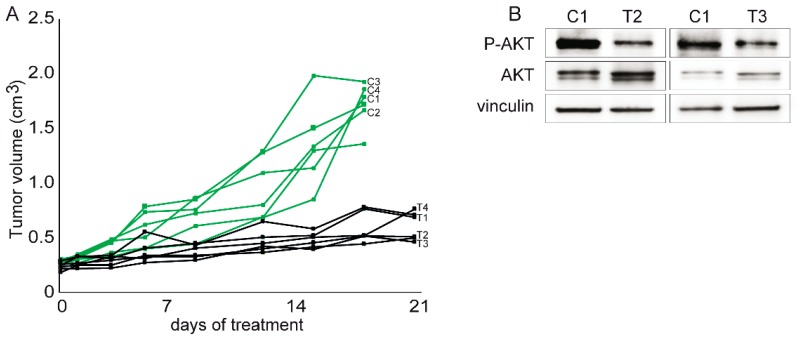
In vivo response of the *PIK3R1*^W624R^ PDXs to buparlisib. Randomized mice were divided into two cohorts and treated with 20 mg/kg buparlisib, administered as described in the Methods section. (**A**) Growth curves of treated (black solid lines) and control (green solid lines) animals. (**B**) Western blot analysis of the phosphorylation of the AKT signal transducer in response to drugs, as a proxy of PI3K activation status in response to drugs, in the individual treated (T2 and T3) and control (C1) PDXs indicated in panel (A).

**Figure 7 cells-09-00442-f007:**
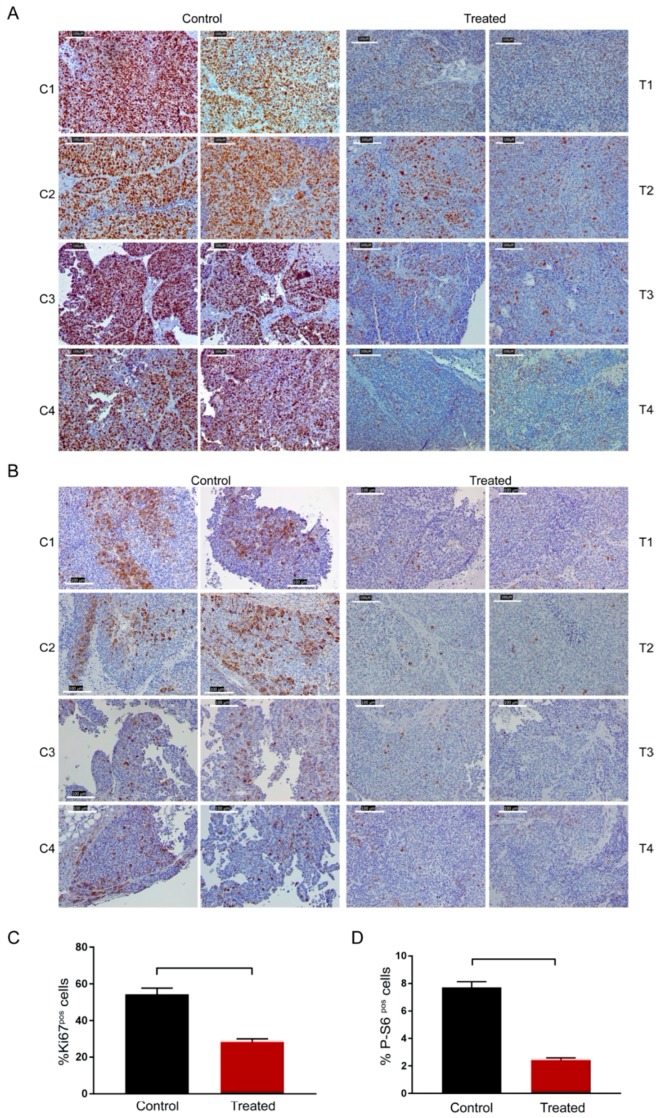
Immunohistochemical detection of proliferation index and decreased activation of PI3K in *PIK3R1*^W624R^ PDXs, treated with buparlisib as shown in Figure 6. (**A**) Representative images of Ki67 positive cells detected in treated and control PDXs grown as shown in Figure 6 panel A; (**B**) representative images of P-S6 positive cells detected in treated and control PDXs grown as shown in Figure 6 panel A. (**C**) Quantification of Ki67 positive nuclei evaluated as a percentage of positive area versus total nuclei area. (**D**) Quantification of P-S6 positive cells evaluated as a percentage of positive area versus total area. The *p* value has been calculated using unpaired *t*-Student test.

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
