# Peer review of "PIK3R1*^W624R^ Is an Actionable Mutation in High Grade Serous Ovarian Carcinoma"

_cells, 2020, doi:10.3390/cells9020442_

Round 1

Reviewer 1 Report

In the manuscript “PIK3R1W624R is an Actionable Mutation in High Grade 3 Serous Ovarian Carcinoma” by D’Ambrosio et al, authors fully characterized 43 lines of PDXs and performed copy number analysis and whole exome sequencing of 12 lines derived from naïve, high-grade EOCs. Authors identified a PIK3R1W624R variant in PDXs from a high grade serous EOC and performed drug response assays on PDX Derived Tumour Cells and in vivo on PDXs.

The manuscript is very clear, but the authors should deepen few points to strengthen their data.

1.      Authors compared susceptibility to PI3K/AKT/mTOR pathway inhibitors of PIK3R1W624R carrying PDX line #475 to that of control cell lines known to be susceptible/resistant to any given drug. Is it feasible to use another PDX line (without any PI3K pathway activating mutation) as a control for those experiments?

2.      Given that it is not possible to clearly define how PIK3R1W624R determines the activation of p110a, is it visible the iperactivation of PI3K pathway in PDX line 475 compared to other PDX lines? Did authors observe an increased basal phosphorylation of Akt in western blot experiments in line 475 compared to not mutated PDX lines? Alternatively, did authors try to perform IHC or IF for pAkt on original tissue #475, compared to other tissues?

3.      In the in vivo experiments, authors observed that buparlisib delayed the growth of the PIK3R1W624R PDXs. Did authors test in vivo also the other inhibitors tested ex vivo? And did authors try the treatment with PI3K inhibitors on PDX derived from not mutated tissues? Moreover, it would have been elegant to detect the decreased phosphorylation of Akt also in IHC analysis on PDX.

4.      Authors described that the PIK3R1W624R PDX line was propagated from a biopsy sample from a patient with a stage IIIc HGS-EOC, that had responded to platinum-based chemotherapy. After relapse, another sample was propagated as PDX line, which did no longer respond to carboplatin. Did authors try to performed the same ex vivo and in vivo experiments also in this paired PDX line?

Author Response

Reviewer 1

Answers point by point. The relevant changes in the revised version are highlighted in yellow

This reviewer asked for the feasibility of using another PDX line (without any PI3K pathway activating mutation) as control in experiments showing the effect of inhibitors of the PI3K/AKT pathway, in addition to cell lines. We did it and have added to the revised version as Supplementary figure S3 the treatment ex vivo with buparlisib of the PDX line #2085, described in Table S1 as derived from a HGS EOC with TP53 and BRCA2 mutation. This control has been quoted in the manuscript test starting from line 294 of the previous version, where also corrections requested by the reviewer no. 2 have been added. As requested by this reviewer we compared the activation status of the PI3K pathway of the PIK3R1W624R mutated PDX line #475 to that of other PDX lines without mutations of the pathway using IHC. As a proxy of PI3K activation in IHC, we used antibodies against the phosphorylated form of the S6 protein, as anti P-AKT antibodies in IHC are less reliable. The newly added Supplementary figure S2 shows the comparison between PDX line #475 and PDX samples showing a high expression of S6 that is however not phosphorylated, as in these PDX lines the PI3K pathway is not genetically affected. This reviewer asked if we have tested PI3K inhibitors other than buparlisib in vivo . We did not, to meet the 3R principles as buparlisib is a pan-class I PI3K inhibitor. He/she also asked if we tried the treatment with PI3K inhibitors on PDX derived from not mutated tissues. As answered to this reviewer’s request at point 1 we have now added experiments performed on the PDX lines #2085 without mutations of genes involved in the PI3K/AKT/mTOR pathway (see above). As requested by this Reviewer we are now showing in the new Fig 7, where also IHC of the Ki67 previously shown in Fig. 6 has been transferred, the decreased phosphorylation of the S6 protein upon treatment in vivo of the mutated PDX line with buparlisib. As mentioned above (see answer to point 2), IHC detection of P-S6 is more reliable than the IHC detection of P-AKT as a proxy of activation of PI3K. This reviewer asked if we have tested PI3K inhibitors on the PDX line propagated from the relapsed EOC of the same patient, that we mentioned in the manuscript, which carries the same PIKR1 gene mutation but did no longer respond to carboplatin. We did not, because any possible result could be inconclusive. If the susceptibility to PI3K inhibition would be maintained we would only further show that this mutation confers the susceptibility. Conversely, if resistance to buparlisib would have occurred, we should have studied the mechanisms of resistance. This latter study could be very complex, possibly inconclusive and exceeding the aim of this paper.

Reviewer 2 Report

In this mansucript, D’Ambrosio and collaborators are interested by identifying cancer drives and actionable mutations. By using a beautiful combination of in vitro and in vivo approaches, the authors identified a variant in PIK3R1 gene in high grade serous ovarian carcinoma.

Overall the manuscript is well written and results are clearly exposed. Although the general objective of the manuscript is interesting and should clearly be usefull to the community, there are a number of points that would be required to be addressed to be considered acceptable for publication.

Cancer drivers and actionable mutation should be better defined in the introduction. The goal of the study in the introduction should be better presented. Fig2 : It is not clear why the authors concluded from this figure « this mutation was predicted to be deleterious and damaging and possibily actionable (Fig.2). Fig4 : did the authors evaluate the possibility to perform immunoprecipritation for p110 and p85 isoforms of PI3K ? this point should be at least discussed. Fig5 : the authors measured cell viability by using the reagent cell titer glo which is known to be more relevant for ATP content an therefor represents an indirect measurement of cell viability. The authors should also evaluate the cell viaility by counting the number of nuclei per well. 5 : the authors did not discuss why they observed a difference in term of ATP content between the different compounds and the mutation. As an example, why did they observe a difference only with dactolisib ? Fig5 : they should also described better the different cell lines used for the functional assay.

Author Response

Reviewer 2

Answers point by point. The relevant changes in the revised version are highlighted in yellow

As requested, definitions of cancer driver and actionable mutations have been added by modifying the Introduction text starting from line 44 of the previous version As requested by this reviewer the goal of the study is better spelt out in the Introduction, at line 51 of the previous version To better clarify the sentence of the line 218 of the previous version and reported by this reviewer « this mutation was predicted to be deleterious and damaging and possibly actionable (Fig. 2)”; we have added a sentence at line 218 of the previous version We did not perform co-immunoprecipitation for p110 and p85 isoforms of PI3K because the amount of p110 and p85 is very small in any tissue. Indeed, these molecules have been identified firstly as enzymatic activity. The presence of minute amount of the proteins of interest makes Co-IP unreliable, in particular when a difference is searched for between binding of p110 to either wt or mutated p85 We fully agree with this reviewer in considering CellTiterGlo results potentially affected by metabolic changes in treated cells. Therefore, rather than merely counting cell nuclei as suggested by the reviewer, we have carried out a crystal violet cell cytostatic/cytotoxicity assay, now added to the manuscript as Supplementary figure S4. This reviewer claimed that experiments with Dactolisib reported in Fig. 5 are not consistent with results obtained with other drugs. We seek him/her to note that Dactolisib has been tested in a larger range of concentrations (from 1nM to 100microM). This because we already know that this type of range was more appropriate for testing this drug. As a result, the curves in panel C of Fig. 5 could have misled the reader. As requested by this reviewer, in the revised version we have described the different cell lines used for the functional assay in the Results section (second last paragraph of the Results section 3.3), while, in the previous version, cell lines were described only in the legend to Fig. 5. The Fig. 5 legend has been shortened accordingly. The same Results paragraph has been changed to take into account the comments of the Reviewer no. 1.

Round 2

Reviewer 1 Report

The authors completely satisfied the review requests.